# Early Cambrian origin of the shelf sediment mixed layer

Romain C. Gougeon[1], M. Gabriela Mángano[1], Luis A. Buatois[1], Guy M. Narbonne[1,2] & Brittany A. Laing[1]

The mixed layer of modern oceans is a zone of fully homogenized sediment resulting from bioturbation. The mixed layer is host to complex biogeochemical cycles that directly impact ecosystem functioning, affecting ocean productivity and marine biodiversity. The timing of origin of the mixed layer has been controversial, with estimates ranging from Cambrian to Silurian, hindering our understanding of biogeochemical cycling and ecosystem dynamics in deep time. Here we report evidence from the Global Stratotype Section and Point (GSSP) of the basal Cambrian in the Burin Peninsula of Newfoundland, Canada, showing that a well-developed mixed layer of similar structure to that of modern marine sediments was established in shallow marine settings by the early Cambrian (approximately 529 million years ago). These findings imply that the benthos significantly contributed to establishing new biogeochemical cycles during the Cambrian explosion.

[1] Department of Geological Sciences, University of Saskatchewan, Saskatoon, SK S7N 5E2, Canada. [2] Department of Geological Sciences and Geological Engineering, Queen's University, Kingston, ON K7L 3N6, Canada. Correspondence and requests for materials should be addressed to R.C.G. (email: gougeon.romain@gmail.com)

The impact of biogenic disturbance on and into the sediment seafloor has captured the interest of many fields, from its influence on geochemical cycles[1–5] and the effect of ecosystem engineers[6–8] to the understanding of factors controlling modern and past vertical zonation of the sediment[9–14]. In modern oceans, sediment below the sediment-water interface is vertically zoned into the mixed layer (approximately uppermost 5–10 cm), the transition layer (up to 20–35 cm) and the deeper historical layer[9–12]. The surface mixed layer comprises fully homogenized sediment resulting from bioturbation by epifaunal and shallow infaunal organisms[9,12,13]. The transitional layer is a heterogeneous zone, characterized by firmer sediment and the activity of suspension and deposit feeders forming open, maintained domiciles and actively infilled structures[10,11]. Below this, the historical layer represents the lowermost zone and is regarded as a relic of previous mixed and transition layers that have

encompassed burial and early lithification process[10]. The presence of the mixed layer dramatically influences substrate stability for benthic animals[11] and biogeochemical cycling between the oceans and recently deposited sediments[9].

In contrast with modern seafloors, Ediacaran marine sediment surfaces were pervasively coated with resistant microbial mats that acted as a geochemical filter between the underlying sediment and overlying seawater[15,16]. Bacteria and microbial communities that build up biofilms secrete extracellular polymeric substances that form bonds between sediment grains which stabilize bedforms[17] and reduce hydraulic conductivity within the sediment[18]. Diffusion of dissolved oxygen from the water column becomes difficult below those mats, resulting in anoxic and sulfidic conditions within the sediment and positioning the redox boundary close to the sediment-water interface[19,20]. Consequently, poor oxygenation of the substrate impacts on

**Fig. 1** Stratigraphic column of the Chapel Island Formation showing the change in degree of sediment mixing. Block diagrams summarize the change in sedimentary fabric and intensity of bioturbation, revealing the establishment of a sediment mixed layer at the base of Cambrian Stage 2. M1 to M5 refers to the five members of the Chapel Island Formation (CIF). Scale is in meters

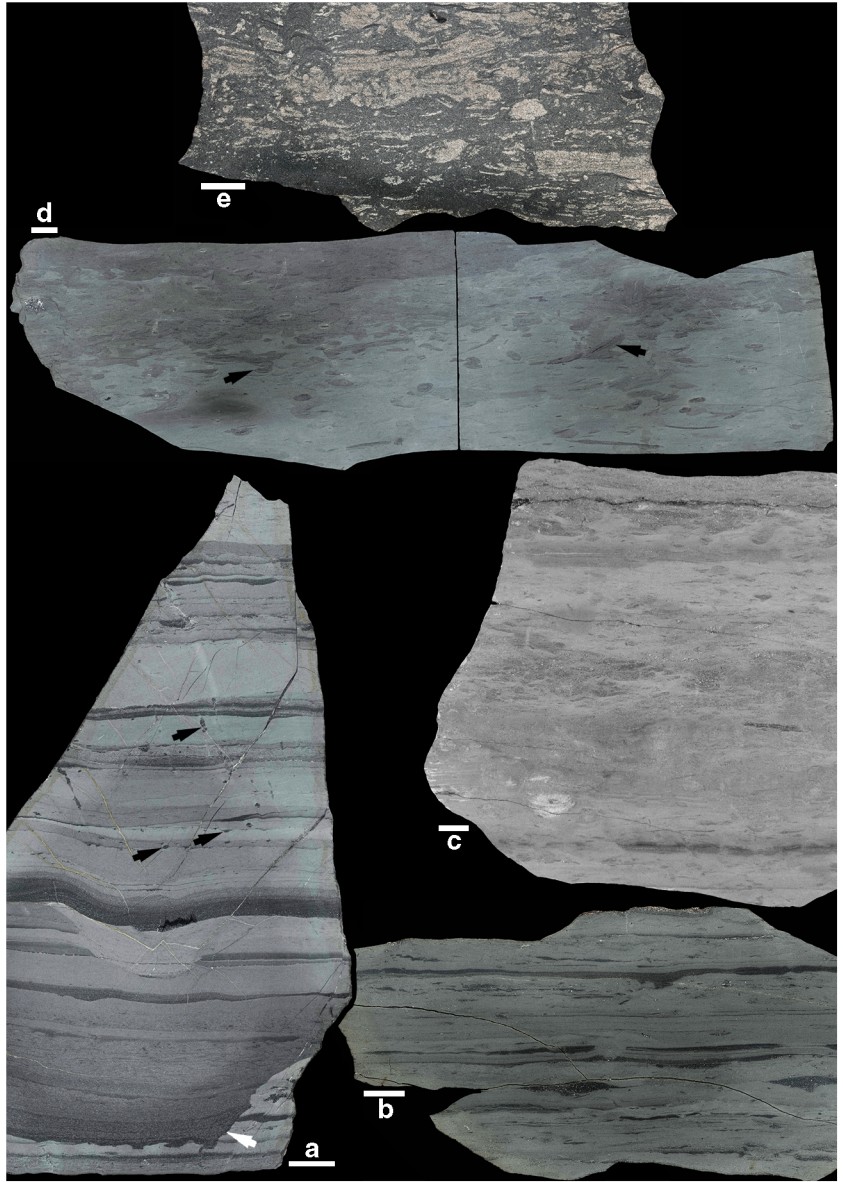

Fig. 2 Polished slabs from the Chapel Island Formation. **a** Heterolithics with well-defined, small discrete burrows (black arrows) and a gutter cast (white arrow), Fortune Head, member 2 A (BI = 0). **b** Heterolithics and mottled textures, Little Dantzic Cove, base of member 3 (BI = 2). **c** Intense mottling and remnant lamination/bedding, Little Dantzic Cove, top of member 3 (BI = 5). **d**, **e** Intense mottled textures overprinted by mid-tier *Teichichnus* (black arrows), Little Dantzic Cove, members 4 (**d**) and 5 (**e**) (BI = 6). See Supplementary Table 1 for detail on BI. Scale bars are 1 cm long

biochemical cycles such as phosphorus[2] and sulfur[1], the latter playing a role in the carbon cycle by acting in the organic matter decomposition[19]. Significant reductions in the prevalence of microbial mats coupled with widespread bioturbation and associated bioirrigation would have permitted increasingly free-interchange between surface sediments and the water column, dramatically changing fluid chemistry of both of these regimes[1,16]. Biomixing would enhance oxidation and reduction processes of ionic elements, while bioirrigation would transport reduced elements from the sediment pore water to the overlying water column where they are therefore oxidized[21]. However, the timing of this transition leading to the establishment of a mixed zone at the sediment-water interface remains uncertain within a 100 million year interval stretching from the early Cambrian to the late Silurian (compare refs [22,23] and [24,25], and references therein), hindering paleontological and geochemical studies of this key milestone in Earth evolution.

Here we provide detailed evidence from the basal Cambrian in the Burin Peninsula of Newfoundland, Canada, demonstrating that a mixed layer of similar structure to that of modern marine sediments one was well established in shallow marine settings by the early Cambrian.

## Results

**Stratigraphic setting.** The type section of the basal Cambrian in the Burin Peninsula of Newfoundland, Canada (Fig. 1, Supplementary Fig. 1) is ideal to unravel the early history of bioturbation. The approximately 1000 m thick Chapel Island Formation (CIF) ranges in age from latest Ediacaran to Cambrian Age 2[26,27] and has been subdivided into five informal members[28] (Fig. 1). Members 1–4 are a continuous succession of fine-grained silici-clastics collectively equivalent to the Quaco Road Member in New Brunswick[29] and are disconformably overlain by member 5,

which is equivalent to the Mystery Lake Member in New Brunswick[29]. Remarkable facies recurrence through this succession (mainly shoreface to offshore[30–32]) permits comparison with similar shallow-marine sedimentary environments through time. In particular, three major outcrops (Fortune Head, Grand Bank Head, and Little Dantzic Cove), each exposing several hundred meters of strata (Fig. 1), provide key information on sedimentary and evolutionary changes through this succession. Of these, Fortune Head is the one that has captured the most attention because it contains the Cambrian Global Stratotype Section and Point (GSSP), which is located 2.4 m above the base of member 2 at the last occurrence of Ediacaran megafossils and the base of the *Treptichnus pedum* ichnofossil Zone[26,27,33–35] (Fig. 1).

**Sediment mixing through the lower Cambrian succession**. Fortunian strata in member 2 consist mainly of nearshore to offshore, very fine-grained to medium-grained sandstone and siltstone with wave and current ripples, convolute and parallel lamination, and locally with gutter casts, pot casts and syneresis cracks[30,32] (Fig. 2a). Event sandstones are pristinely preserved showing well-defined, sharp bases and tops. Trace fossils occur as isolated and discrete burrows, either attached to the overlying sandy bed (adhering preservation style) or unattached and surrounded by a silty matrix (floating preservation style)[36]. Bioturbation Index (BI of Taylor and Goldring[37]) is predominantly 0–2 (up to 30% disturbance). Burrows consist of sharp, shallow-tier, 1–2 cm deep treptichnids and tiny *Palaeophycus*, and 5–6 cm deeper *Gyrolithes*, all being interpreted as open burrows of worm-like organisms passively filled by the next sedimentation event[36,38]. This preservation style (i.e., isolated, sharp-walled, and passively filled open burrows) suggests firmground conditions close or at the sediment surface in the absence of a fully homogenized layer of mixed sediment. In addition, these strata show a wide variety of microbially induced sedimentary structures associated with vermiform grazing trails and arthropod scratch marks, further implying a stabilized substrate[39]. These strata display evidence for considerably more penetration of the seafloor by burrowing organisms than in Ediacaran strata lower in the section and worldwide, but Fortunian burrows are discrete and do not form a mixed layer[22,24,36,39,40].

Member 3 shows similar trace fossil preservation, but with a higher bioturbation index that is typically 2 and reaches 5 in some beds (Figs. 1, 2b). In these mainly offshore deposits, the horizontal lamination is overprinted by the shallow-tier to mid-tier feeding structure *Teichichnus* (a horizontal burrow displaying vertical displacement and produced by either worms or arthropods) with the large-size, mollusk-like grazing trail *Psammichnites* preserved on bedding planes. The latter first appears sparingly at the base of the *Rusophycus avalonensis* ichnofossil Zone[33] (Fig. 1), but does not become abundant until member 3. The upper interval of member 3 and all of member 4 show, for the first time, layers where the primary fabric is totally disrupted by *Teichichnus* overprinting an indistinct burrow mottling background fabric, resulting in a completely homogenized sediment (BI = 6), revealing the presence of a well-developed mixed layer (Figs. 1, 2c, 2d). This change in sedimentary fabric is essentially coincident with the first appearance of abundant small shelly fossils in the succession (*Watsonella crosbyi* zone), marking the base of Cambrian Stage 2 that forms the upper half of the Terreneuvian Series[26].

Member 5 is a sandstone-dominated succession that thickens, coarsens, and shallows upward from offshore to shoreface deposits[31]. Despite the increase of hydrodynamic energy associated with shallowing, sharp-based event sandstones have diffuse tops and contain discrete *Teichichnus* which cross-cut the primary lamination, passing upwards into intensely bioturbated storm sandstone and fair-weather mudstone that preserve only locally remnants of the primary lamination (BI = 5). The tendency towards an increasing homogenization of the sediment is confirmed by the middle strata of member 5 (Figs. 1, 2e), which show full biogenic disturbance leaving no vestiges of primary lamination (BI = 6). Finally, the top of tempestites from the upper interval of member 5 contain deep-tier, U-shaped *Diplocraterion*, showing suspension-feeding infauna that were able to colonize sandy deposits in the immediate aftermath of storm events.

Analysis of the CIF shows that substrate utilization shifted from non-penetrative, simple horizontal trails largely restricted to grazing the microbial mats (Ediacaran), to a mixture of mat grazing trails and undermat-mining burrows and small penetrative burrows yielding a firmground fabric (Fortunian), to the first appearance of complete bioturbation and the establishment of a well-developed mixed layer (Cambrian Stage 2). These changes are best displayed in deposits that accumulated between fair-weather and storm wave base (i.e., offshore and offshore transition), but are also evident in deposits formed immediately above fair-weather wave base (i.e., lower shoreface) and below storm wave base (i.e., shelf), underscoring the environmental extent of the mixed layer across the depositional gradient.

## Discussion

The results of our study represent a sharp departure from recent work in the area, which suggested that no increase in degree of bioturbation is apparent through the Cambrian portion of the CIF[35]. Also, previous ichnologic data from the CIF supporting the notion that sediment mixing was insignificant during the early Cambrian were based on field observations from an 18 m-thick interval near the base of the Fortunian[24,41] and, therefore, are not representative of the whole unit. Inferences of global suppression of sediment mixing were based on measurements of degree of bioturbation restricted to heterolithic facies from a number of sections spanning Cambrian-Silurian strata[24,25,41]. This methodological approach to the study of sediment mixing is problematic because heterolithic facies are defined by intercalations of discrete layers of sandstone and mudstone, therefore necessarily implying low to moderate intensities of bioturbation that allow preservation of the primary fabric[23]. In contrast, ichnofabric analysis of polished slabs of fair-weather mudstone samples in this study reveals intense bioturbation and the establishment of a well-developed and relatively thick mixed layer of typical Phanerozoic structure.

The fossil record of bioturbated sediments consists of stacked historical layers comprising discrete burrows emplaced in the transition layer overprinting undifferentiated bioturbation mottling that typifies the mixed layer[10]. Vertical accretion of the seafloor results in the upward migration of the infaunal communities. As a result, and in contrast to modern seas where a snapshot view of the vertical partitioning of the infaunal habitat is available, it is not possible to accurately measure the thickness of the mixed layer in the fossil record (with the notable exception of frozen tiered profiles[11]). This suggests that previous studies implying that the mean sediment mixed layer thickness was 0.2 cm in early and middle Cambrian (with a maximum thickness of 0.5 cm)[41] are hard to support on conceptual and empirical grounds. Although no precise value for the thickness of the mixed layer can be provided, the fact that the Cambrian Stage 2 marine facies of the CIF show discrete burrows sharply overprinting a mottled background fabric in identical fashion to modern sediments is evidence of a well-developed mixed layer.

Our detailed study in this continuous succession supports the hypothesis that the Cambrian explosion involved two phases[22]. The Fortunian phase was characterized by the appearance of a wide variety of behavioral strategies as revealed by an increase in diversity and disparity of trace fossils accompanied by increase exploitation of microbial mats[22,39]. The Cambrian Stage 2 phase is characterized by an increase in depth and extent of bioturbation that was conducive to the establishment of the mixed layer and the onset of a true Phanerozoic ecology. The abundance of trace fossils of sediment bulldozers and shallow-tier to mid-tier feeding structures severely impacted on the sedimentary fabric and its bioirrigation, representing a new phase in ecosystem engineering[22,42]. In addition to increased bioturbation in mud-dominated and silt-dominated settings below fair-weather wave base, nearshore sand-dominated settings experienced a remarkable increase in extent and depth of bioturbation by Cambrian Age 2 as evidenced by colonization of mobile sandy substrates by a suspension-feeder infauna. Establishment of this infauna may have dramatically affected geochemical cycles by increasing regeneration of nitrogen and phosphorous to the water column, enhancing fluxes of organic carbon and dissolved inorganic nitrogen into the sediment[22,42]. Intense bioturbation changed the circulation of nutrients and oxygen into the seafloor by increasing the irrigation levels, resulting in the downward shift of the redox boundary and leading to colonization of this new ecological niche[22,43,44]. Because the intensity of bioturbation highly depends on the food supply (i.e., the flux of carbon arriving in the sediment)[13], the activity of new bioturbators as ecosystem engineers could have impacted the carbon cycle by increasing the provision and use of fresh organic matter in deeper levels of the sediment. This may have resulted in a positive feedback on biodiversity, increasing the utilizable ecospace, with microbial biomass developing deeper into the sediment[6]. Metabolic, bacterial, and meiofaunal activities are often enhanced in the zone immediately surrounding tubes and burrows[44]. Modern studies show that grazing and deposit feeding strongly influence the supply of ammonium into the sediment necessary for the growth of microphytobenthos and thus enhance primary productivity[44,45]. Rapid changes in the carbon cycle are in concomitance with important pulses of metazoan diversification during the early Cambrian, the most notable being close to the base of the Cambrian Stage 2[46].

Ichnologic evidence from Avalonia shows that the mixed layer was established over a 12 million year interval of early Cambrian time. This key event in ecosystem engineering[6,7] would have profoundly affected the physical and chemical characteristics of the substrate, decreasing its cohesion and bearing capacity while increasing its irrigation and oxygenation. These changes both required and permitted major evolutionary adaptations of organisms living on and in the Cambrian seafloor, contributing to the rapid evolutionary feedback of the Cambrian explosion. Adaptation by organisms to this seafloor turnover resulted in larger body size, and an increase in mobility and density of individuals[6]. Extinctions, adaptations, and environmental restriction of benthic communities were the ecological results of the "Cambrian Substrate Revolution"[23,47]. For instance, a well-bioturbated sediment seafloor impacted on echinoderms, contributing to the extinction of helicoplacoids[48] and forcing edrioasteroids to change their mode of attachment in order to colonize harder substrates[47,49]. Recent studies imply that most Cambrian echinoderms were attached to skeletal debris, therefore being pre-adapted to colonize hardgrounds in the Furongian[50]. Moreover, many modern invertebrate clades use their mitochondria or chemosymbionts in their tissues to oxidize sulfide which is highly abundant and toxic below microbially stabilized seafloors[19], an adaptation that can be traced back at least to the

Fortunian. The significance of bioturbation as a driving force of ecosystem functioning is consistent with information documenting the establishment of complex food webs[51,52] and infaunal tiered communities[42] during the early Cambrian. The opening of new ecological niches triggered by the establishment of infaunal bioturbators further supports geochemical studies documenting a rise of oxygenation in the oceans by Cambrian Age 2[2,53,54]. However, the Cambrian Age 2 increase in bioturbation may have resulted in higher rates of oxidation of organic matter, therefore reducing carbon burial, lowering atmospheric $O_2$, and increasing ocean anoxia, which may have triggered a subsequent decrease in biodiversity[2]. Additional studies exploring in detail the body-fossil and trace-fossil record are needed to test this geobiologic hypothesis.

Our results show that a mixed layer was established in the Avalon region of the Iapetus Ocean during Cambrian Age 2. Further work is required to confirm how geographically extensive the development of the mixed layer in early Cambrian oceans was or if significant diachronism among paleocontinents was involved in infaunalization[55]. However, the presence of intensely bioturbated deposits of the same age in Mongolia[42,56] suggests establishment of a mixed layer in shallow-marine settings outside the realm of the Iapetus Ocean, potentially pointing towards a more global phenomenon.

## Methods

**Polishing of samples and computer processing**. Forty-seven rock samples were collected at regular intervals in Fortune Head, Grand Bank Head, and Little Dantzic Cove stratigraphic sections, parallel to a sedimentologic bed-by-bed analysis. For the purpose of this study only 36 samples are illustrated herein (five in Fig. 2, 31 in Supplementary Notes 1–7). Polishing in the remaining 11 samples was not considered satisfactory due to the presence of pervasive microfractures.

All of the collected samples were cut and polished at the University of Saskatchewan (Canada), revealing their vertical internal organization of physical and biogenic sedimentary structures. We implemented the technique developed by Dorador et al.[57] and Dorador and Rodríguez-Tovar[58]. After scanning/photographing of the polished slabs, each sample was digitally improved by following the same protocol in Adobe Photoshop: (a) adjustment of the levels of the image, which increase the differences between pixels by stretching the histogram of pixel values; (b) adjustment of the contrasts/brightness, which controls the amount of light and improve the tone differences; and (c) adjustment of the vibrance, which controls the yellow tones and turn the image to less artificial grey tones. Each of the 36 samples was processed by these three adjustments uniformly on the image.

Bioturbation Indexes (sensu Taylor and Goldring[37]) were evaluated after attribution of a bioturbation percentage using Adobe Photoshop and ImageJ softwares. By following the method of Cao et al.[59], this process consists of (a) delineating the bioturbated area of each slab surface using the "lasso tool" in Adobe Photoshop and painting it in black; (b) measuring the "total area" and "bioturbated area" numbers (in pixels) using the "wand tool" in ImageJ; and (c) calculating the bioturbation percentage in order to precisely identify the bioturbation index of the sample.

**Bioturbation index**. Ichnological observations have been framed by using a combined ichnofacies and ichnofabric approach. Accordingly, ichnological parameters analyzed include ichnotaxonomic composition, ethologic and trophic types, degree of bioturbation, and tiering structure. In depth discussions of all these parameters are presented elsewhere[42]. Degree of bioturbation has been measured in the field and for each polished slab by assessing the percentage of primary sedimentary fabric that has been affected by biogenic activity. Subsequently, we have assigned a bioturbation index or BI following a widely utilized previous scheme[37], based on a previous scale[60], comprising seven categories. BI = 0 indicates no bioturbation (0%), and a pristine primary fabric. BI = 1 (1–4%) is represented by sparse bioturbation with few discrete trace fossils locally overprinting the well-preserved primary fabric. BI = 2 (5–30%) characterizes low bioturbation in sediment having distinct, well-preserved sedimentary structures. BI = 3 (31–60%) is typified by discrete trace fossils, moderate bioturbation and relatively sharp bedding boundaries. BI = 4 (61–90%) indicates intense bioturbation, high density and common overlap of trace fossils, and primary sedimentary structures that have been for the most part erased. BI = 5 (91–99%) represents sediment with completely disturbed bedding and intense bioturbation. BI = 6 (100%) illustrates fully bioturbated and totally reworked sediment, as a result of repeated emplacement of biogenic structures. Tiering classification is based on a previous scheme[22], which

considers a subdivision in shallow tier (less than 6 cm), mid-tier (6–12 cm), deep tier (12–100 cm), and very deep (more than 100 cm).

**Environmental typology**. Environmental subdivisions to frame ichnological data are based on the following scheme[42,61–64]. The shoreface extends from the low-tide line to fair-weather wave-base, and can be subdivided into the upper, middle, and lower shoreface. The upper shoreface is subjected to multidirectional current flows in the build-up and surf zones, and its deposits consist of trough and planar cross-stratified, well-sorted, coarse-grained to medium-grained sandstone. The middle shoreface occurs in the area of shoaling and initial breaking of waves, and its deposits include swaley cross-stratified, trough cross-stratified, and combined-flow ripple cross-laminated, well-sorted, medium-grained to fine-grained sandstone. The lower shoreface is located immediately above fair-weather wave base, and its deposits are represented by amalgamated, thick hummocky cross-stratified fine-grained to very fine-grained sandstone that locally contain wave and combined-flow ripples. Millimetric mudstone partings locally occur between some hummocky cross-stratified units.

The offshore is defined as the zone between the fair-weather wave base and the storm wave base, and can be subdivided into the offshore transition and the upper and lower offshore. The offshore transition is most proximal region of the offshore and occurs right below the fair-weather wave base. Offshore-transition deposits are typically represented by regularly interbedded, parallel-laminated to burrowed mudstone, and thin to thick erosive-based, fine-grained to very fine-grained sandstone with hummocky cross-stratification, and combined-flow and wave ripples at the top. The upper offshore is present between the offshore transition and the lower offshore. Upper-offshore deposits consist of mudstone intervals interbedded with thin, laterally extensive, erosionally based, very fine-grained silty sandstone layers that may contain parallel lamination, hummocky cross-stratification, combined-flow ripples, and wave ripples. The lower offshore is placed immediately above the storm wave base. Lower-offshore deposits are mudstone-dominated, but they may contain laterally-extensive, sharp-based, erosive storm-emplaced, very fine-grained silty sandstone with combined-flow ripples and parallel lamination.

The shelf is located right below the storm wave base, extending to the slope break. Shelf deposits typically consist of mudstone locally interbedded with thin normally graded siltstone and very fine-grained sandstone layers.

**Data availability**. The authors declare that the data supporting the findings of this study are available within the paper and its supplementary information files.

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

## Acknowledgements

Financial support for this study was provided by Natural Sciences and Engineering Research Council (NSERC) Discovery Grants 311727–15, 311726–13, and 05561-2014 awarded to Mángano, Buatois, and Narbonne respectively. We are grateful to B. Novakovski for his help on sample processing. We appreciate the feedback of Matthew Clapham on an early draft of this manuscript.

## Author contributions

R.C.G. and B.A.L. collected the samples in the field; R.C.G. polished, cut and prepared the samples; M.G.M., L.A.B., and G.M.N. supervised the project. All authors wrote the manuscript.

## Additional information

**Competing interests:** The authors declare no competing interests.

