## [Peer Review File · Nature Communications]

Reviewers' comments:

Reviewer #1 (Remarks to the Author):

Dear Mr. Gougeon and co-workers,

These are my comments concerning your paper entitled "Early Cambrian origin of the sediment mixed layer" submitted to Nature Communications with reference number NATCOMMS-17-39174.

This is a really important contribution demonstrating that the mixed layer is Cambrian in origin. The paper is in general well written and most conclusions are fully supported by reported data. The origin of the mixed layer in the Cambrian is a debated issue, and recent paper by Tarhan et al. 2015 suggests that it was not until the Silurian when the mixed layer was fully developed. By contrast, Gougeon et al. uses primary data on bioturbation from a classic locality in the Burin Peninsula (Newfoundland) to demonstrate that the mixed layer was already developed in the early Cambrian.

The mixed layer is an important condition of modern marine environments in which bioturbation increases the water content of surficial layers of sediment and blurred the sediment-water interface. The transition from a typical Proterozoic-type substrate without intense bioturbation to a mixed layer has had a major impact on the ecology and evolution of many non-burrowing benthic metazoans. This major event has been linked to the so-called Cambrian Substrate Revolution (CSR) (Bottjer et al., 2000; Bottjer, 2010). There are several papers that used polish section to demonstrate that Cambrian rocks show little evidence of bioturbation, like those from the Cambrian Series 2 Poleta Formation (Dornbos and Bottjer, 2000) and the Cambrian Series 3 Crisholm Shale (Domke and Dornbos, 2010), both in Laurentia. In several parts of the paper (including title and line 190), Gougeon et al. suggest that the origin of a mixed layer was probably a global phenomenon; but those aforementioned evidences point to the opposite situation. What I think is really important is the fact that Gougeon et al. clearly demonstrate that the origin of the mixed layer was Cambrian; whether its origin was a global phenomenon or not is something that requires further information from all palaeocontinents, and ideally samples collected from different environments and the same age.

In the line 36 the authors claim that "the presence of the mixed layer dramatically influences substrate stability for benthic animals". Later, in line 176 they claim that "these changes both required and permitted major evolutionary adaptations of organisms living on and in the Cambrian sea floor". I really think that it is important for the reader discuss these ideas a little more. There are several papers (see citations above) suggesting that some typical Cambrian groups (for example suspension feeding echinoderms) contributed importantly to the Cambrian Substrate Revolution, and that the increase in bioturbation during the Cambrian forced those groups to adopt new strategies. For contrast a recent paper by Zamora et al., 2017 suggested that only some edrioasteroids were adapted in the Cambrian to a typical stabilized substrate lacking a mixed layer. Could you expand your discussion indicating some examples of how the Cambrian origin of the mixed layer permitted major adaptations in some metazoan groups?

Finally, I just want to congratulate for your interesting paper and important conclusions that clearly demonstrate the mixed layer had a Cambrian origin. Your paper is a clear example of how careful collecting of primary data in the field and lab can contribute resolving major evolutionary questions.

Sincerely yours,

References:

Bottjer, D.J., 2010, The Cambrian substrate revolution and early evolution of the phyla: Journal of

Earth Science, v. 21, p. 21–24.

Bottjer, D.J., Hagadorn, J.W., and Dornbos, S.Q., 2000, The Cambrian substrate revolution: GSA Today, v. 10, p. 1–7.

Domke, K.L., and Dornbos, S.Q., 2010, Paleoecology of the middle Cambrian edrioasteroid echinoderm *Totiglobus*: Implications for unusual Cambrian morphologies— Palaios, v. 25, p. 209–214.

Dornbos, S.Q., and Bottjer, D.J., 2000, Evolutionary paleoecology of the earliest echinoderms: Helicoplacoids and the Cambrian substrate revolution: Geology, v. 28, p. 839–842.

Zamora, S., Deline, B., Álvaro, J. J., & Rahman, I. A. 2017. The Cambrian Substrate Revolution and the early evolution of attachment in suspension-feeding echinoderms. Earth-Science Reviews, 171, 478-491.

Reviewer #2 (Remarks to the Author):

The start of the Cambrian defines a pivotal moment in the co-evolution of life and the geosphere, as at this very moment, animal life evolved near the seafloor. Yet, more and more evidence has recently accumulated that this emergence of animal life also changed the way our planet biogeochemically functions (by impacting the geochemical cycles of carbon and sulphur, and hence changing atmospheric levels of CO₂ and O₂). The impact of animal evolution on global biogeochemical cycling occurs through bioturbation (the mixing of sediments by burrowing fauna), and both the timing as well as the intensity of this “bioturbation feedback” is the subject of an important (and sometimes controversial) ongoing debate. Accordingly, any paper that makes an important progress on the subject is of great interest to the journal.

The present manuscript provides an important contribution to this ongoing debate; based on an excellent dataset a clear conclusion is presented. While previous publications have argued for a protracted evolution of mixing, the current manuscript argues exactly the opposite: a mixed sediment layer is already present early on in the Cambrian, indicating that substantial burrowing occurred early on in the evolution of animal life. The authors attribute their different conclusion to a methodological bias in previous studies (which analyze heterolithic facies, compared to the polished slabs of homolithic mudstone that are analyzed here).

I think this manuscript has the right quality and potential impact to deserve publication. However, I have one important reservation. The entire discussion of the environmental context and biogeochemical implications (which encompasses a major part of the discussion) is based on isotope data, which are not presented here. Instead the reader is referred to a single reference, which turns out to be a non-peer reviewed conference abstract (ref 2). In my view, the isotope data should be included in the manuscript, but I leave it up to the judgement of the editor to decide on this.

Additional comments and suggestions:

Title. This work deals with shallow sediments (above the storm wave base). A similar response in deeper waters is currently unknown. So I suggest to make this explicit in the title. “The Early Cambrian origin of the mixed layer in shelf sediments”

L14. Non-cohesive-> incorrect Terminology -> Muddy sediments are still considered cohesive, despite being bioturbated

L16 Ecosystem performance -> this is not correct terminology in ecology -> Ecosystem functioning

L22 "fully marine settings" What is meant by this? Instead it is adamant to mention that this work only deals with shallow sediments, and that a similar response in the deep ocean remains to be seen

L26. Ref 2 is a conference abstract (not peer reviewed). Moreover these opening sentences would benefit from referencing review papers that discuss bioturbation in contemporary marine sediments and the resulting effects on geochemistry. For example:

Meysman, F. J. R., Middelburg, J. J., and Heip C. H. R. (2006) Bioturbation: a fresh look at Darwin's last idea. *Trends in Ecology and Evolution* 21, 688-695.

Aller R. C. (2001) Transport and Reactions in the Bioirrigated Zone. In *The Benthic Boundary Layer* (ed. B. P. Boudreau and B. B. Jorgensen). pp. 269-301. Oxford University Press.

Maire O., Lecroart P., Meysman F., Rosenberg R., Duchêne J., Grémare A. (2008) Quantification of sediment reworking rates in bioturbation research: a review. *Aquatic Biology* 2, 219-238.

L39 "seal" and L41 "free-interchange". This conceptual image of a microbial mat as a seal is not entirely correct. To my knowledge, the cited references 1 and 12 have also never used this conceptual image. A seal to what? H₂S? Both non-bioturbated and bioturbated sediments do not leak H₂S (as H₂S is also oxidized in the oxygenated burrows). Rather, microbial mats act as a geochemical filter for mineralization products produced in deeper sediments (catalyzing anabolic nutrient uptake and metabolic redox transformations)

L54 Add a reference to Figure 1

L67 It is confusing that member 2 is subdivided in Fig 2 into M2a and M2b, while this separation is not used or discussed in the manuscript.

L67 mention that "offshore" and similar terms are exactly defined in the methods

L73 predominantly 0-2 -> this is confusing, as the Upper Fortune in Fig 2 also mentions BI = 4
L147 increased

L151 "increased matground penetration" Penetration of what? Moreover, "increased matground penetration" does not provide a mechanism for the increased oxidation of sulfur and carbon. The increased oxidation of reduced carbon and sulfur compounds, is linked to transport of reduced solids upwards (by biomixing) and the transport of O₂ and nitrate downward (by bio-irrigation).

L155 "sediment consistency" This is not a standard term in sediment geochemistry. What does it mean? Porosity? Cohesiveness?

L162 "increasing ocean ventilation" This is nonsensical. Ocean ventilation is a physical process driven by thermohaline circulation. How can this be influenced by bioturbation?

L169 "carbon cycle, resulting in positive feedback on biodiversity" Biodiversity in the sediment or water column? And how does this feedback work? Bioturbation increases oxidation of organic matter, reduces carbon burial, lowers atmospheric O₂, increases ocean anoxia, and this will generally decrease biodiversity.

L174. In the abstract and here, one references an age model. This age model is however not shown in Fig 2 (or the other figures)

L179 Ecosystem performance -> this is not correct terminology in ecology -> Ecosystem functioning

L180 trophic webs -> incorrect ecological term -> food webs

L182 I cannot see how "our study further supports geochemical studies documenting a rise of oxygenation in the oceans" As explained above, bioturbation would promote anoxia, not increased oxygenation.

L189 delete "of other terranes"

L193 sedimentary ecosystem functioning

L207 the same pattern on -> the same protocol in

L212 "on the image surface"-> what does this mean??

L239 Environmental typology

Responses to the Reviewers

We thank the two reviewers for their support for our conclusions, and for requesting clarifications on our wording and elaboration on the implications of our conclusions. Reactions to the points raised in the reviews are listed below.

Reviewer #1:

“In the line 36 the authors claim that “the presence of the mixed layer dramatically influences substrate stability for benthic animals”. Later, in line 176 they claim that “these changes both required and permitted major evolutionary adaptations of organisms living on and in the Cambrian sea floor”. I really think that it is important for the reader discuss these ideas a little more. There are several papers suggesting that some typical Cambrian groups (for example suspension feeding echinoderms) contributed importantly to the Cambrian Substrate Revolution, and that the increase in bioturbation during the Cambrian forced those groups to adopt new strategies. For contrast a recent paper by Zamora et al., 2017 suggested that only some edrioasteroids were adapted in the Cambrian to a typical stabilized substrate lacking a mixed layer. Could you expand your discussion indicating some examples of how the Cambrian origin of the mixed layer permitted major adaptations in some metazoan groups?”

The impact of the formation of the mixed layer on the benthos during the early Cambrian is now emphasized in the following sentences that were added in the discussion:

Adaptation by organisms to this seafloor turnover resulted in larger body size, and an increase in mobility and density of individuals⁶. Extinctions, adaptations, and environmental restriction of benthic communities were the ecological results of the “Cambrian substrate Revolution”^{40,46}. For instance, a well-bioturbated sediment seafloor impacted on echinoderms, contributing to the extinction of helicoplacoids⁴⁷ and forcing edrioasteroids to change their mode of attachment in order to colonize harder substrates^{46,48}. Recent studies imply that most Cambrian echinoderms were attached to skeletal debris, therefore being pre-adapted to colonize hardgrounds in the Furongian⁴⁹. Moreover, many modern invertebrates are able to oxidize sulfide, which is highly abundant and toxic below microbially stabilized seafloors¹⁹, an adaptation that probably appeared during the Fortunian.

Reviewer #2:

“Title. This work deals with shallow sediments (above the storm wave base). A similar response in deeper waters is currently unknown. So I suggest to make this explicit in the title. “The Early Cambrian origin of the mixed layer in shelf sediments”

Agreed. The title of the manuscript was modified as follow:

Early Cambrian origin of the shelf sediment mixed layer

“L14. Non-cohesive-> incorrect Terminology -> Muddy sediments are still considered cohesive, despite being bioturbated”

Agreed. The term “non-cohesive” was removed.

“L16 Ecosystem performance -> this is not correct terminology in ecology -> Ecosystem functioning”

Agreed. This correction was accepted.

“L22 “fully marine settings” What is meant by this? Instead it is adamant to mention that this work only deals with shallow sediments, and that a similar response in the deep ocean remains to be seen”

“Fully marine settings” is used in its traditional sense to distinguish these deposits from those of alluvial, river, brackish, continental, or lacustrine settings.

We have further clarified the shallow marine condition, as follows:

Here we report evidence from the Global Stratotype Section and Point (GSSP) of the basal Cambrian in the Burin Peninsula of Newfoundland, Canada, showing that a well-developed mixed layer of similar structure to that of modern marine sediments was established in shallow marine settings by the early Cambrian (approximately 529 my ago).

“L26. Ref 2 is a conference abstract (not peer reviewed). Moreover these opening sentences would benefit from referencing review papers that discuss bioturbation in contemporary marine sediments and the resulting effects on geochemistry. For example:

Meysman, F. J. R., Middelburg, J. J., and Heip C. H. R. (2006) Bioturbation: a fresh look at Darwin's last idea. *Trends in Ecology and Evolution* 21, 688-695.

Aller R. C. (2001) Transport and Reactions in the Bioirrigated Zone. In *The Benthic Boundary Layer* (ed. B. P. Boudreau and B. B. Jorgensen). pp. 269-301. Oxford University Press.

Maire O., Lecroart P., Meysman F., Rosenberg R., Duchêne J., Grémare A. (2008) Quantification of sediment reworking rates in bioturbation research: a review. *Aquatic Biology* 2, 219-238.”

Agreed. Former Reference 2 was included in three places in our previous submission to give credit to ongoing work by a separate research group in the same section as our study (peer-reviewed papers still have not been produced). The geochemical dataset that is the base of their ISECT 2017 abstract is distinct from our study and will be published separately by their team. We believe that adding a complex set of geochemical data from their paper would distract from rather than elucidate our analysis based on systematic sampling and polished slabs of early Cambrian sea floor. References to their abstract have been replaced with references to full papers on early Cambrian geochemistry (e.g. Maloof et al., 2010) and the references on the impact of bioturbation suggested by Reviewer 2. We note that our brief discussion of geochemistry is fully consistent with previously published studies, most notably the one by Boyle et al. (2014) which is referenced in our manuscript.

Our manuscript has been altered by adding references in the opening sentence:

The impact of biogenic disturbance on and into the sediment seafloor has captured the interest of many fields, from its influence on geochemical cycles¹⁻⁵ and the effect of ecosystem engineers⁶⁻⁸ to the understanding of factors controlling modern and past vertical zonation of the sediment⁹⁻¹⁴.

“L39 “seal” and L41 “free-interchange”. This conceptual image of a microbial mat as a seal is not entirely correct. To my knowledge, the cited references 1 and 12 have also never used this conceptual image. A seal to what? H₂S? Both non-bioturbate and bioturbated sediments do not leak H₂S (as H₂S is also oxidized in the oxygenated burrows). Rather, microbial mats act as a geochemical filter for mineralization products produced in deeper sediments (catalyzing anabolic nutrient uptake and metabolic redox transformations)”

We think the reviewer meant Refs. 11 and 12, as these two are the ones cited regarding this point. Refs. 11 and 12 do mention in fact that mats act as seals. The “seal” is mainly referring to the diffusion of oxygen into the sediment from the water column, placing the redox boundary close to the sediment water interface and thus preventing oxygenation of the interstitial waters. For clarification, we have replaced the word “seal” by “geochemical filter” as requested, and have added the following sentences on the nature of this filter:

Bacteria and microbial communities that build up biofilms secrete extracellular polymeric substances that form bonds between sediment grains which stabilize bedforms¹⁷ and reduce hydraulic conductivity within the sediment¹⁸. Diffusion of dissolved oxygen from the water column becomes difficult below those mats, resulting in anoxic and sulfidic conditions within the sediment and positioning the redox boundary close to the sediment-water interface^{19,20}. Consequently, poor oxygenation of the substrate impacts on biochemical cycles such as phosphorus² and sulfur¹, the latter playing a role in the carbon cycle by acting in the organic matter decomposition¹⁹.

“L54 Add a reference to Figure 1”

Agreed. This reference to Fig. 1 was added.

“L67 It is confusing that member 2 is subdivided in Fig 2 into M2a and M2b, while this separation is not used or discussed in the manuscript.”

Agreed. We removed the subdivision in Member 2A and 2B from Figure 1.

“L67 mention that “offshore” and similar terms are exactly defined in the methods”

As explained in the Methods, the terms used for the subdivision of shallow-marine environments are used in other peer-reviewed papers in the same way. Detailed definitions are provided in the Methods section and we think further precisions are not necessary earlier in the text.

“L73 predominantly 0-2 -> this is confusing, as the Upper Fortune in Fig 2 also mentions BI = 4”

Agreed. Figure 2 was modified to redress that issue.

“L147 increased”

Agreed and changed accordingly.

“L151 “increased matground penetration” Penetration of what? Moreover, “increased matground penetration” does not provide a mechanism for the increased oxidation of sulfur and carbon. The increased oxidation of reduced carbon and sulfur compounds, is linked to transport of reduced solids upwards (by biomixing) and the transport of O₂ and nitrate downward (by bio-irrigation).”

We were referring to the penetration of mats by burrowers which increases the free exchange between the water column and the sediment. The mechanism explaining increased oxidation within the sediment is now outlined in the text, in connection with the comment on the sealing capacity of mats (see previous correction on lines 39 and 41).

Similar details are then given in the paragraph about the mat sealing action, as well as in the 3rd paragraph of the discussion (line 152 to 179 in the revised manuscript; see also comment below about line 169 issue).

“L155 “sediment consistency” This is not a standard term in sediment geochemistry. What does it mean? Porosity? Cohesiveness?”

Agreed. The sentence was modified as follows:

The abundance of trace fossils of sediment bulldozers and shallow- to mid-tier feeding structures dramatically affected the sedimentary fabric and its bio-irrigation, representing a new phase in ecosystem engineering^{21,41}.

“L162 “increasing ocean ventilation” This is nonsensical. Ocean ventilation is a physical process driven by thermohaline circulation. How can this be influenced by bioturbation?”

Agreed. The last part of the sentence was removed.

L169 “carbon cycle, resulting in positive feedback on biodiversity” Biodiversity in the sediment or water column? And how does this feedback work? Bioturbation increases oxidation of organic matter, reduces carbon burial, lowers atmospheric O₂, increases ocean anoxia, and this will generally decrease biodiversity.

Here, we refer to the role of bioturbation in deepening the redox continuity surface and how it may have promoted colonization by other bioturbators. This part has been rewritten and enhanced to clarify that point as follows:

Because the intensity of bioturbation highly depends on the food supply (i.e. the flux of carbon arriving in the sediment)¹³, the activity of new bioturbators as ecosystem engineers could have impacted the carbon cycle by increasing the provision and use of organic matter in the sediment. This may have resulted in a positive feedback on biodiversity, increasing the utilizable ecospace, with microbial biomass developing deeper into the sediment⁶. Metabolic, bacterial and meiofaunal activities are often enhanced in the zone

immediately surrounding tubes and burrows⁴³. Modern studies show that grazing and deposit feeding strongly influence the supply of ammonium into the sediment necessary for the growth of microphytobenthos and thus enhance primary productivity^{43,44}. Rapid changes in the carbon cycle are in concomitance with important pulses of metazoan diversification during the early Cambrian, the most notable being close to the base of the Cambrian Stage 2⁴⁵.

In addition, we have introduced later in the text a sentence outlining the point made by the reviewer (lines 200-204):

However, the Cambrian Age 2 increase in bioturbation may have resulted in higher rates of oxidation of organic matter, therefore reducing carbon burial, lowering atmospheric O₂, and increasing ocean anoxia, which may have triggered a subsequent decrease in biodiversity². Additional studies exploring in detail the body- and trace-fossil record are needed to test this geobiologic hypothesis.

“L174. In the abstract and here, one references an age model. This age model is however not shown in Fig 2 (or the other figures)”

Ages are indicated in Figure 1. These are based on classic work done in the section, where zonations were based on biostratigraphic information (e.g. small shelly fossils).

“L179 Ecosystem performance -> this is not correct terminology in ecology -> Ecosystem functioning”

Agreed and changed accordingly.

“L180 trophic webs -> incorrect ecological term -> food webs”

Agreed and changed accordingly.

“L182 I cannot see how “our study further supports geochemical studies documenting a rise of oxygenation in the oceans” As explained above, bioturbation would promote anoxia, not increased oxygenation.”

We have modified this sentence as follows:

The opening of new ecological niches triggered by the establishment of infaunal bioturbators further supports geochemical studies documenting a rise of oxygenation in the oceans by Cambrian Age 2^{2,52,53}. However, the Cambrian Age 2 increase in bioturbation may have resulted in higher rates of oxidation of organic matter, therefore reducing carbon burial, lowering atmospheric O₂, and increasing ocean anoxia, which may have triggered a subsequent decrease in biodiversity². Additional studies exploring in detail the body- and trace-fossil record are needed to test this geobiologic hypothesis.

“L189 delete “of other terranes”

Agreed and changed accordingly.

“L193 sedimentary ecosystem functioning”

Agreed and changed accordingly.

L207 the same pattern on -> the same protocol in

Agreed and changed accordingly.

L212 “on the image surface”-> what does this mean??

Agreed. The term “surface” was removed to clarify the idea.

“L239 Environmental typology”

Agreed and changed accordingly.

REVIEWERS' COMMENTS:

Reviewer #1 (Remarks to the Author):

Authors have followed all my suggestions and the resulted MS has included several discussions that satisfied my requirements pointed out in the first review.

Reviewer #2 (Remarks to the Author):

The manuscript has been thoroughly revised and improved following the suggestions of the reviewers. I recommend publication provided that three remaining issues are addressed.

[1] The authors have added a new section describing the geochemical changes during the Cambrian Substrate Revolution. They suggest that suggest that the most prominent driver of these changes was a change in the oxygen penetration depth of the sediment (L42 to L50):

"Diffusion of dissolved oxygen from the water column becomes difficult below those mats, resulting in anoxic and sulfidic conditions within the sediment and positioning the redox boundary close to the sediment-water interface. Consequently, poor oxygenation of the substrate impacts on biochemical cycles such as phosphorus and sulfur, the latter playing a role in the carbon cycle by acting in the organic matter decomposition. Significant reductions in the prevalence of microbial mats coupled with widespread bioturbation would have permitted increasingly free interchange between surface sediments and the water column, dramatically changing fluid chemistry of both of these regimes"

This is a simplistic (and in my view highly mistaken) view of the geochemical changes during the Cambrian Substrate Revolution.

We recently investigated this transition in detail by comparing a state-of-the-art sediment diagenetic model with and without biomixing (solid phase transport) and bio-irrigation (van de Velde and Meysman (2016) Aquatic Geochemistry, DOI 10.1007/s10498-016-9301-7).

This study convincingly shows the opposite of the above claim: the advent of biomixing results in an important decrease (rather than an increase) in the oxygen penetration depth in the sediment. Accordingly, the prominent changes in sediment geochemistry at the transition are not due to changes in the oxygen penetration depth.

Effectively, the most prominent change in geochemical cycling during the Cambrian Substrate Revolution is the drastic increase in the rate of the redox cycling in the sediment (van de Velde and Meysman, 2016)

(a) Biomixing dramatically increases the reoxidation of reduced compounds, and hence stimulates the redox cycling of sulphur and iron in the sediment, and frees the sediment from free sulfide (toxic to marine benthos)

(b) Bio-irrigation results in an increase of the oxic volume in the sediment (via the creation of oxygenated burrows linings) and a substantially increases in the surface area of redox interfaces stimulating redox cycling

[2] Line 172. The bioturbating ecosystem engineers do not change the overall flux of organics to the sediment (this is set by primary production and export in the overlying water). They do inject fresh organics into deeper layers. therefore, I suggest to change (L171-172)

the provision and use of organic matter in the sediment.

To:

the provision and use of fresh organic matter in deeper sediment strata.

[3] Line 193

"Moreover, many modern invertebrate clades are able to oxidize sulfide"

This is not correct. The invertebrates do not oxidize sulfide themselves, but have sulfide oxidizing bacteria as symbionts.

Sincerely, Filip Meysman

Responses to Reviewers

Reviewer #1:

We thank the referee for positive comments on our manuscript and the indication that it is now suitable for publication.

Reviewer #2:

“[1] The authors have added a new section describing the geochemical changes during the Cambrian Substrate Revolution. They suggest that the most prominent driver of these changes was a change in the oxygen penetration depth of the sediment (L42 to L50):

"Diffusion of dissolved oxygen from the water column becomes difficult below those mats, resulting in anoxic and sulfidic conditions within the sediment and positioning the redox boundary close to the sediment-water interface. Consequently, poor oxygenation of the substrate impacts on biochemical cycles such as phosphorus and sulfur, the latter playing a role in the carbon cycle by acting in the organic matter decomposition. Significant reductions in the prevalence of microbial mats coupled with widespread bioturbation would have permitted increasingly free interchange between surface sediments and the water column, dramatically changing fluid chemistry of both of these regimes"

This is a simplistic (and in my view highly mistaken) view of the geochemical changes during the Cambrian Substrate Revolution. We recently investigated this transition in detail by comparing a state-of-the-art sediment diagenetic model with and without biomixing (solid phase transport) and bio-irrigation (van de Velde and Meysman (2016) Aquatic Geochemistry, DOI 10.1007/s10498-016-9301-7). This study convincingly shows the opposite of the above claim: the advent of biomixing results in an important decrease (rather than an increase) in the oxygen penetration depth in the sediment. Accordingly, the prominent changes in sediment geochemistry at the transition are not due to changes in the oxygen penetration depth.

Effectively, the most prominent change in geochemical cycling during the Cambrian Substrate Revolution is the drastic increase in the rate of the redox cycling in the sediment (van de Velde and Meysman, 2016):

(a) Biomixing dramatically increases the reoxidation of reduced compounds, and hence stimulates the redox cycling of sulphur and iron in the sediment, and frees the sediment from free sulfide (toxic to marine benthos)

(b) Bio-irrigation results in an increase of the oxic volume in the sediment (via the creation of oxygenated burrows linings) and a substantially increases in the surface area of redox interfaces stimulating redox cycling”

Our study is on bioturbation, not the details of the mechanisms in the oxidation of ionic elements which affects geochemical cycles. We thank the referee for his comments, and have added a short discussion on the effects of bioirrigation and the reference he requested, as follows:

“Significant reductions in the prevalence of microbial mats coupled with widespread bioturbation and associated bioirrigation would have permitted increasingly free-interchange between surface sediments and the water column, dramatically changing fluid chemistry of both of these regimes^{1,16}. Biomixing would enhance oxidation and reduction processes of ionic elements, while bioirrigation would transport reduced elements from the sediment pore water to the overlying water column where they are therefore oxidized²¹.”

“[2] Line 172. The bioturbating ecosystem engineers do not change the overall flux of organics to the sediment (this is set by primary production and export in the overlying water). They do inject fresh organics into deeper layers. Therefore, I suggest to change (L171-172) “the provision and use of organic matter in the sediment.” to “the provision and use of fresh organic matter in deeper sediment strata.”

Agreed. This correction was accepted (although we have rephrased the term “sediment strata” which is technically not correct).

“[3] Line 193: “Moreover, many modern invertebrate clades are able to oxidize sulfide”. This is not correct. The invertebrates do not oxidize sulfide themselves, but have sulfide oxidizing bacteria as symbionts.”

The referee’s statement is only partly correct – some marine invertebrates use symbiotic algae to process the hydrogen sulfide but others are able to use their own mitochondria to partly or completely process hydrogen sulfide. For example, Volkel and Grieshaber (1997) demonstrated that the lugworm *Arenicola marina* is able to oxidize sulfide in its mitochondria as an energy source. It is true that some organisms use symbiotic bacteria to provide a food source from sulfide oxidation like in the clam *Solemya reidi*, but the first steps of oxidation here occur again in mitochondria from gills and foot tissues (Powell and Somero, 1986). Similar reports have been made from other invertebrates: e.g. the California killifish *Fundulus parvipinnis* and the speckled sanddab *Citharichthys stigmaeus* (Bagarinao and Vetter, 1990); the priapulida *Halicryptus spinulosus* (Oescher and Vetter, 1992); the polychaete *Heteromastus filiformis* (Oescher and Vismann, 1994).

Other readers might have similar queries, so we have modified our sentence in the text as follows:

“Moreover, many modern invertebrate clades use their mitochondria or chemosymbionts in their tissues to oxidize sulfide which is highly abundant and toxic below microbially stabilized seafloors¹⁹, an adaptation that can be traced back at least to the Fortunian.”